# Effect of the Notch a Cursor Cannot Enter for Pointing Movement Time

Yosuke Oba*
Meiji University

Homei Miyashita†
Meiji University

## Abstract

The notch on the top edge of the MacBook Pro (2021) display hides the mouse cursor even though the cursor can move under this area. Avoiding the notch or moving the cursor carefully around the notch can increase the movement time. In this study, we perform three experiments to evaluate the effect of the notch on the movement of the mouse cursor. In Experiment 1, we showed that the notch increases the pointing movement time under specific scenarios. In Experiment 2, we showed that it is better to avoid the notch instead of moving the cursor under the notch given its current specification. Finally, in Experiment 3, we showed that changing the notch to an area where the cursor cannot enter is an effective approach that allows the user to point at the target more rapidly and accurately if the target is adjacent to the notch. This is because the outer edge of the notch stops the cursor, and this results in faster and more accurate target pointing. Thus, **the notch should be an area where the cursor cannot enter**.

**Keywords:** Graphic user interface, mouse pointing, human motor performance, notch, edge target, Fitts' law

**Index Terms:** H.5.2 [User Interfaces]: User Interfaces—Graphical user interfaces (GUI);

## 1 Introduction

A notch is used to position the web camera in a display for increasing the usable area of the display. For example, the MacBook Pro (2021) has a notch at the center of the top edge on the display (Figure 1). It is the black area on the display that cannot be used; however, the cursor can enter this area and is hidden partially or entirely when the cursor enters the notch.

Pointing, i.e., using the cursor to point at targets such as buttons or icons, should be fast and accurate. Two factors that affect the movement time are target size and distance from the initial position of the cursor to the target [11, 19]. The movement time increases with an increase in distance and a decrease in target size. Further, placing distractors (which do not hide the cursor) on the path to the target increases the movement time [6, 25]. The notch can cause a user to miss the cursor position when it is inside the notch or to lose sight of the cursor, which can increase movement time. Avoiding the notch or moving the cursor carefully around the notch can increase the movement time. We performed three experiments to evaluate the effect of the notch on the movement of the mouse cursor. In Experiment 1, we recorded longer movement times because the cursor was hidden by the notch when moving between targets at the top edge of the display. In this experiment, participants moved the cursor according to two strategies in the current specification of the notch (an area where a cursor can enter): The first strategy was to enter the notch and move the cursor along the top edge of the screen (along-strategy); the second strategy was to avoid the notch (avoid-strategy). The results also showed that the closer the target is to the notch, the longer the movement time. In Experiment

*e-mail: bonscow@gmail.com
†e-mail: homei@homei.com

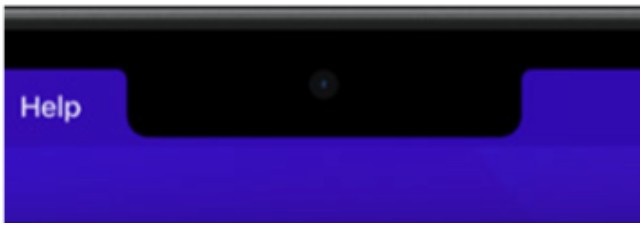

Figure 1: Notch on the MacBook Pro (2021). The screen is not displayed in the black area. Although the cursor can enter this area, it is hidden partially or entirely by the notch. Thus, a user can easily lose sight of the cursor.

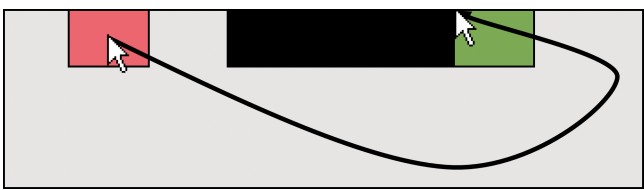

Figure 2: The expected path of the cursor when the notch is changed to an area where the cursor cannot enter.

2, we investigated which of these strategies is preferred given the current specification of the notch. The results of this experiment indicated that the movement time was shorter when using the avoid-strategy compared to that using the along-strategy. Because the avoid-strategy was found to be desirable, changing the notch to an area where the cursor cannot enter can be considered. In Experiment 3, we changed the specification of the notch to an area where the cursor cannot enter. The results of this experiment indicated that changing the notch to an area where the cursor cannot enter allowed faster and more accurate pointing at the target that is adjacent to the notch.

Based on these results, this study suggests that the notch should be an area where the cursor cannot enter. Fast and accurate pointing of a target adjacent to the notch can be achieved when the notch does not hide the cursor and the cursor stops at the outer edge of the notch (Figure 2).

## 2 Related Work

### 2.1 Effect of Losing the Cursor

Patrick et al. found out that a user loses the cursor when using multiple displays with different resolutions based on an unnatural cursor movement between displays, and proposed a Mouse Ether technique [5]. The proposed technique improved performance by up to 28% by preventing unnatural warping when the cursor was moved between displays.

Hollinworth et al. found that senior citizens lose the cursor because of poor eyesight and sustained concentration, and therefore, they implemented a Field Mouse (a mouse with a touch sensor attached) and proposed a technique wherein the cursor moves to the center of the screen when the user holds the mouse [15]. This technique help reduce the time required to search for the cursor, which

in turn reduces the movement time.

Stephane et al. focused on screen torus settings [16]. With this setting, when the cursor reaches the screen edge, it appears from the opposite end. For example, when the cursor reaches the right edge, it appears from the left edge. However, users can easily lose sight of the cursor when it warps around the edges. To overcome this issue, they proposed a TorusDesktop technique that adds appropriate visual feedback between the time the cursor warps.

These studies focused on the user losing sight of the cursor, but these did not focus on a scenario where the cursor is hidden. To our knowledge, no study has investigated a scenario where the cursor becomes invisible. The scenario where the cursor disappears is similar to one where user loses the cursor, and therefore, a similar trend may be observed.

## 2.2 Performance Models on Pointing Motions

Fitts' law [11, 19] can help predict the movement time ($MT$) based on the index of difficulty ($ID$), which is determined by the distance from the initial position of the cursor to the target center ($A$) and target width ($W$). Shannon formulation is widely used in human–computer interaction (HCI) research. It is expressed as

$$MT = a + bID, \; ID = \log_2 \left( \frac{A}{W} + 1 \right) \qquad (1)$$

where $a$ and $b$ represent empirical constants. Hereinafter, $a$, $b$, $b_1$, $b_2$, and $c$ also represent empirical constants.

Target height ($H$) also affects the movement time because typical targets on graphical user interfaces (GUIs) are rectangular [3, 8, 14, 20, 22, 30]. Accot and Zhai [1] proposed a model for a bivariate (2D) pointing tasks that considers $H$. Further, Zhang et al. [31] proposed to balancing the effects of $W$ and $H$.

$$MT = a + b\log_2 \left( \sqrt{c \left( \frac{A}{W} \right)^2 + (1-c) \left( \frac{A}{H} \right)^2} + 1 \right) \qquad (2)$$

Yamanaka [28] showed that the movement time for pointing to an edge target from another edge target on the same edge can be predicted by Eq 2.

The effect of distractors (e.g., buttons and icons that users do not want to point at) placed near the target for the movement time has been investigated. Blanch et al. [6] showed that movement time decreases with a decrease in distractor density. However, the model proposed by Blanch et al. assumed a scenario where distractors and the target $ID$s are equal. Usuba et al. [25] showed that the movement time increases with a decrease in the interval between the target and the distractor under a scenario where the $ID$s of the distractor and target are different. Further, they proposed a model that considers the interval ($I$) between the target and the distractor.

$$MT = a + b_1 \log_2 \left( \sqrt{\left( \frac{A}{W_{click}} \right)^2 + c \left( \frac{A}{W_{visual}} \right)^2} + 1 \right) + b_2 \log_2 \left( \frac{1}{I + 0.0049} + 1 \right) \quad (3)$$

where $W_{click}$ and $W_{visual}$ represent the clickable width and visual width of the target, respectively. When $W_{click}$ and $W_{visual}$ are equal, Eq. 3 can be approximated to Eq 4.

$$MT = a + b_1 \log_2 \left( \frac{A}{W} + 1 \right) + b_2 \log_2 \left( \frac{1}{I + 0.0049} + 1 \right) \quad (4)$$

Jax et al. [17] proposed a model for the case when there is a distractor on the path to the target and the user avoids the distractor.

$$MT = a + b\log_2 \left( \frac{A}{W} + 1 \right) + cB \qquad (5)$$

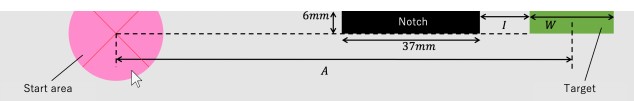

Figure 3: Schematic of Experiment 1 where $Position = Inside$ and $Start = Far$.

Vaughan et al. noted an increase in the empirical constants for the Jax et al. model.

$$MT = a + b\log_2 \left( \frac{A + 2B}{W} + 1 \right) \qquad (6)$$

where $B$ represents the minimum distance required to avoid the distractor and is perpendicular to the straight path to the target. For consistency in Eq. 1, Eq. 5 and Eq. 6 were modified to correspond to Eq. 1. Yamanaka et al. [29] studied the effect of the distractor on movement time in crossing.

However, these studies focused on distractors that do not hide the cursor. To our knowledge, distractors that hide the cursor (such as the notch) have not been investigated.

## 2.3 Pointing Operations at Screen Edges

An edge target (target adjacent to the edge of the screen) can reduce the movement time [3, 9, 10, 27, 28]. Pointing at a target at the center of the screen requires the cursor to stop inside the target. The cursor stops at the edge when pointing at an edge target. Thus, the pointing task can be completed by moving a cursor horizontally relative to the edge to which the target is adjacent. Further, a target adjacent to the corner of the screen can be pointed at fast simply by hitting the corner with the cursor [27].

## 2.4 Path Efficiency

A pointing operation for an edge target exploits the fact that the cursor stops at the edge of the screen to complete the pointing without precise control. However, pushing-edge behavior, i.e., pushing the cursor to the edge of the screen, increases the distance traveled by the mouse, and this increases the movement time. Yamanaka [28] defined $PE$ (Path Efficiency) to calculate the efficiency of the cursor movements (Eq. 7).

$$PE = \frac{\text{on-screen travel distance}}{\text{virtual travel distance}} \times 100\% \qquad (7)$$

According to Eq. 7, $PE$ is calculated from the total distance, which includes the virtual path (virtual travel distance) and the on-screen path (on-screen travel distance). A lower $PE$ indicates that the user exhibits a higher pushing-edge behavior, which implies the cursor path is less efficient. A higher $PE$ indicates that the user focuses on shortening the off-screen travel distance instead of pushing the cursor across the edge of the screen. In this case, the user moves the mouse more carefully than is necessary. Note that a $PE$ value close to 100% does not always mean that the movement time is shortened.

## 3 EXPERIMENT 1

Placing distractors that do not hide the cursor on the path to the target can increases the movement time [6, 25]. With the notch, the user can miss the cursor position inside the notch or lose sight of the cursor, which can increase movement time because the user could try to avoid the notch or move the mouse cursor carefully near the notch. In Experiment 1, we investigated the effect of the notch on movement time.

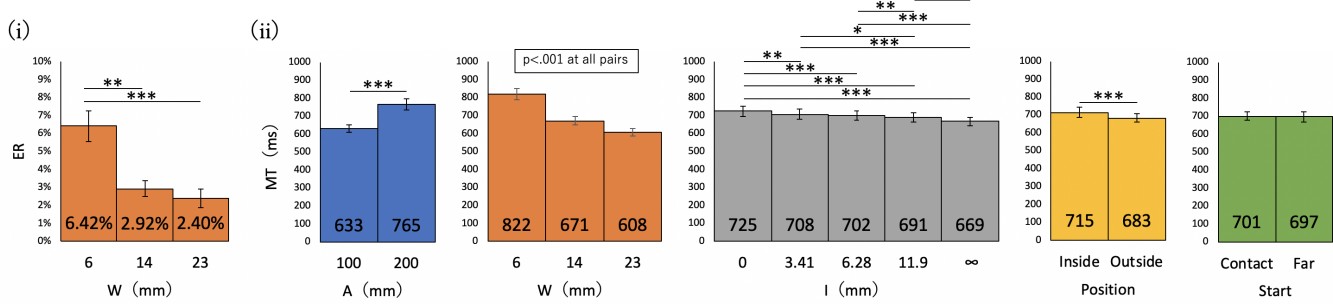

Figure 4: Results of (i) *ER* for *W*. (ii) *MT* for each parameter.

## 3.1 Apparatus

We used a Dell XPS 13 (Intel Core i7-6700, 3.4 GHz; 16 GB RAM; Windows 10 Home). A display was manufactured by ASUS (model VZ249HR; 23.8" diagonal, 1920×1080 pixels) and its refresh rate was set at 75 Hz. We used an optical mouse (Logitech gaming mouse, G-PPD-002WLr; 1600 DPI, and the mouse-cursor speed based on the OS setting was set to the middle of the slider in the control display and the "*Enhance pointer precision*" setting was turned on to match the usual settings of the participant.). The experimental system was implemented with Hot Soup Processor 3.6 and used in the full-screen mode[1].

## 3.2 Participants

A total of 12 local university students participated in the experiment. The average age was 21.8 years ($SD = 2.33$); all participants were experienced mouse user and used their dominant hand (right hand).

## 3.3 Task

The task was created by referring to a previous study [28]. Figure 3 shows a schematic of the task. A pink circular start area (251-pixel radius) and a green target were displayed on a gray background. The participants clicked on the start area; the cursor positioned at the center of the start area. We strictly fixed the starting position of the cursor for the trial assuming that the initial position of the cursor can affect the cursor path and performance of pointing [28]. The trial started once the participant clicked on the starting position. The start area then disappeared, which acted as feedback to indicate the start of the trial. Participants aimed at the target and ended the trial with the next click. If participants clicked the target correctly, we marked the trial as a success; else, the trial was marked as a failure (error). We presented a sound feedback in response to the success or failure of the trial.

## 3.4 Design

The notch was set as a black rectangle (height = 6 mm (22 pixels); width = 37 mm (135 pixels)) based on the size of the notch on the MacBook Pro (2021).

The start area was a circle with a diameter of 251 pixels (68.8 mm); we drew a red line crossed by an X to indicate center of the circle at the starting position of the trial. We set the starting position of the cursor (*Start*) as the condition [28]. *Start = Contact* indicates that the starting position was adjacent to the top edge of the screen. *Start = Far* indicates that the starting position was 6 mm (22 pixels) (the same as the notch height) away from the top edge of the screen.

The target was a rectangle with a height of 6 mm (22 pixels), the same as the notch height, and a width (*W*) of 6, 14, and 23 mm (22,

51, and 84 pixels). We set the *W* values based on the actual sizes of the targets placed on the taskbar of MacBook Pro (2021).

The movement amplitude (*A*) from the starting position to the center of the target was 100 and 200 mm (364 and 729 pixels). We set the *A* values such that the interval between *ID*s in Eq. 1 was approximately constant.

The interval (*I*) between the notch and the target was set as 0, 3.41, 6.28, 11.9, and ∞ mm (0, 12, 23, 44, and ∞ pixels). We set the *I* values with reference to the values of the previous study that investigated the interval between the distractor and the target [25]. Note that an interval of ∞ mm indicates a condition with no notch.

We defined the notch position (*Position*) as the condition. *Position = Inside* indicates that the notch is placed between the start area and the target, and *Position = Outside* indicates that the notch is placed to the left of the target. An equivalent effect is observed at angles of entry that are lineally symmetric about the y-axis when the angle of entry the target adjacent to a top edge with respect to the target is based on the y-axis [3]. Therefore, the performance is the same whether the target is to the left or right of the starting area. We always place the starting area to the left of the target to avoid increasing the workload of the participant.

No significant difference was observed in movement time between a common arrow cursor and a circle shaped cursor when aiming at a target at the top of the screen [3]. We used the common arrow cursor and set the cursor size to be the same size as the actual size set by default on the MacBook Pro (2021).

## 3.5 Procedure

One *set* comprises a random ordering of $2(A) \times 3(W) \times 5(I) \times 2(Start) \times 2(Position) = 120$ conditions. First, the participants practiced 20 trials randomly selected from these conditions. Then, the participants performed 10 sets (1200 trials) for data collection. After completing 5 sets, the participants took a 2 min break. Each participant took approximately 40 min to complete the experiment.

We instructed the participants to (1) point the target as quickly and accurately as possible after clicking the starting position, (2) avoid any clutching action (floating the mouse in the middle of an operation) during the trial, and (3) check the presented conditions before starting the trial. The clutching action decreases the model fit of Fitts' law [7]. We also instructed the participants to avoid the clutching action to restrict the effect for the model that fits the experimental conditions. *A* was set the distance where the cursor could be moved without a clutching action; no participant performed the clutching action during the trial.

## 3.6 Measurements

*MT* represents the time from clicking the start area to clicking the target, excluding the trials with errors. *ER* represents the percentage of clicks made outside the targets.

---

[1]We uploaded the source code and data to `https://research.miyashita.com/papers/I47`

# 4 RESULTS OF EXPERIMENT 1

We recorded 14,400 ($2(A) \times 3(W) \times 5(I) \times 2(Start) \times 2(Position) \times 10(set) \times 12(participants)$) trials including 563 error trials ($ER = 3.91\%$). This error rate was close to that reported in the previous study [28] (3.54%). We analyzed only error-free trials by *mean-of-means* calculation using repeated-measures ANOVA with a Bonferroni post-hoc test because we used the same analysis methods as in the previous studies [25, 28] and because ANOVA is robust against violations of normality [21]. Error bars in the graphs show standard errors; ***, **, and * indicate $p < 0.001$, $p < 0.01$, and $p < 0.05$, respectively.

## 4.1 Error Rate (*ER*)

We observed the main effect of $W$ ($F_{2,22} = 25.3$, $p < 0.001$, $\eta_p^2 = 0.967$) (Figure 4 (i)). The pair-wise comparisons showed that error rates increased with a decrease in $W$. The other parameters did not show the main effects. No significant interaction was observed.

## 4.2 Movement Time (*MT*)

Figure 4 (ii) shows the results of *MT*. We observed the main effects of $A$ ($F_{1,11} = 115$, $p < 0.001$, $\eta_p^2 = 0.913$), $W$ ($F_{2,22} = 375$, $p < 0.001$, $\eta_p^2 = 0.971$), $I$ ($F_{4,44} = 46.1$, $p < 0.001$, $\eta_p^2 = 0.0756$), and *Position* ($F_{1,11} = 30.8$, $p < 0.001$, $\eta_p^2 = 0.737$). The pair-wise comparisons showed that the *MT* for *Position* = *Inside* was longer than that for *Position* = *Outside* ($p < 0.001$). Further, *MT* increased with an increase in $A$ ($p < 0.001$), decrease in $W$ ($p < 0.001$ for all pairs), and decrease in $I$ ($I = 0 \times I = 3.41$ ($p = 0.00506$), $I = 0 \times I = 6.28$ ($p < 0.001$), $I = 0 \times I = 11.9$ ($p < 0.001$), $I = 0 \times I = \infty$ ($p < 0.001$), $I = 3.41 \times I = 11.9$ ($p = 0.0293$), $I = 3.41 \times I = \infty$ ($p < 0.001$), $I = 6.28 \times I = 11.9$ ($p = 0.00359$), $I = 6.28 \times I = \infty$ ($p < 0.001$), and $I = 11.9 \times I = \infty$ ($p = 0.0330$)). We observed significant interactions of $A \times I$ ($F_{4,44} = 4.17$, $p = 0.00599$, $\eta_p^2 = 0.275$), $W \times I$ ($F_{8,88} = 6.71$, $p < 0.001$, $\eta_p^2 = 0.379$), $W \times Start$ ($F_{2,22} = 3.93$, $p = 0.0345$, $\eta_p^2 = 0.264$), $I \times Position$ ($F_{4,44} = 8.25$, $p < 0.001$, $\eta_p^2 = 0.430$), $Start \times Position$ ($F_{1,11} = 6.57$, $p = 0.0264$, $\eta_p^2 = 0.374$), $A \times Start \times Position$ ($F_{1,11} = 7.99$, $p = 0.0165$, $\eta_p^2 = 0.421$), and $A \times W \times I \times Position$ ($F_{8,88} = 2.56$, $p = 0.0148$, $\eta_p^2 = 0.189$).

## 4.3 Path Efficiency (*PE*)

For all participants, *PE* ranged from 98.5%–99.9%. We observed the main effects of $A$ ($F_{1,11} = 9.41$, $p = 0.0107$, $\eta_p^2 = 0.461$), $W$ ($F_{2,22} = 8.03$, $p = 0.00240$, $\eta_p^2 = 0.422$), and *Start* ($F_{1,11} = 47.401$, $p < 0.001$, $\eta_p^2 = 0.812$). The pair-wise comparisons showed that the *PE* for *Start* = *Far* is larger than that for *Start* = *Contact* ($p < 0.001$), and $W = 6$ is the biggest ($W = 6 \times W = 14$ ($p = 0.0272$), and $W = 6 \times W = 23$ ($p = 0.0421$)). The *PE* increased with an increase in $A$ ($p = 0.0107$). These results on $A$, $W$, and *Start* were consistent with those reported in a previous study [28]. We observed significant interactions of $A \times W$ ($F_{2,22} = 5.09$, $p = 0.0152$, $\eta_p^2 = 0.316$) and $A \times Start$ ($F_{1,11} = 84.6$, $p < 0.001$, $\eta_p^2 = 0.885$).

# 5 DISCUSSION OF EXPERIMENT 1

## 5.1 Effects of *I* and *Position* for Movement Time

There was a main effect on the interval ($I$) between the notch and the target, and we found a significant difference from no notch condition ($I = \infty$) for the other conditions ($I = 0, 3.41, 6.28$, and $11.9$). These results indicate that placing the notch increases the movement time. Moreover, movement time increases with a decrease in $I$, which indicates that placing the target near the notch increases movement time.

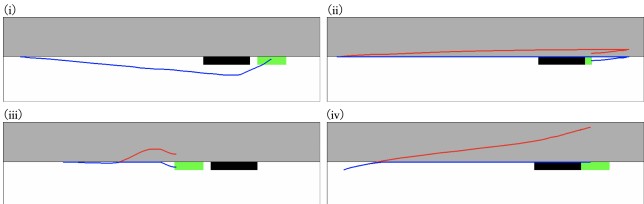

Figure 5: Example cursor trajectories. The red lines indicate the virtual cursor trajectories assuming that there is no screen edge. The blue lines indicates the actual on-screen trajectories.

Table 1: Model fitting for candidate models.

| Model | $a$ | $b_1$ | $b_2$ | $c$ | adj. $R^2$ | AIC |
|---|---|---|---|---|---|---|
| Eq. 1 | 231 | 129 | | | 0.919 | 587 |
| Eq. 2 | 152 | 145 | | 0.956 | 0.927 | 583 |
| Eq. 4 | 223 | 129 | 4.56 | | 0.932 | 579 |
| Eq. 8 | 143 | 145 | 4.56 | 0.956 | 0.941 | 574 |

Another effect was observed, in that *Position* = *Inside* had a longer movement time than that for *Position* = *Outside*. We observed a significant interaction of $I \times Position$. At $I = 0$, the movement time increased compared to that for the condition of no notch by approximately 11.8% in *Position* = *Inside*, and by approximately 4.93% in *Position* = *Outside*.

## 5.2 User Strategy

All participants pointed out that the notch affected their mouse operations. For example, they often lost track of the cursor position within the notch. In *Position* = *Inside*, three participants answered that they avoided the notch (Figure 5 (i)) so that the cursor would not be hidden by the notch. Three other participants answered that they intentionally moved the cursor so that it passed the notch by a large margin (Figure 5 (ii)). Such strategies resulted in longer paths and increased movement time.

In *Position* = *Outside*, three participants stated that they slowed down the cursor early in the movement (Figure 5 (iii)) to avoid the cursor from being hidden by the notch as it passed the target. Although the cursor was not hidden by the notch because of the early deceleration, this strategy also increased movement time.

Four participants answered that they intentionally used edges (Figure 5 (iv)) in all conditions. However, the notch hid the cursor, which caused them to lose sight of the cursor.

All participants answered that they attempted to find the cursor by moving the mouse vigorously when the cursor was hidden by the notch. Participants had to move the mouse to find the cursor because the notch hid the cursor, and this increased movement time.

## 5.3 Model Fitting

Table 1 lists the results of model fitting using all 120 data points. We showed the Akaike information criterion (*AIC*) values because of the different number of constants included in the model along with the adjusted $R^2$ data [2]. A model with a higher adj. $R^2$ and lower *AIC* was defined as the better model. When the difference between the *AIC*s was higher than 2, the difference was worth considering; when it was higher than 10, it was considered significant. In Experiment 1, we set the starting position of the trials, and we excluded Eqs. 5 and 6 from the comparison because the values of $B$ in these models could not be obtained correctly.

Similar to Yamanaka's results [28], Fitts' law (Eq. 1) showed good fits above the typical threshold ($R^2 > 0.9$ [23]). Zhang et al.'s model (Eq. 2) showed a higher adj. $R^2$ and a lower *AIC* than Fitts' law. Usuba et al.'s model (Eq. 4) that considered the interval between

the target and distractor showed a higher adj. $R^2$ and a lower *AIC* than those of Zhang et al.'s model.

Usuba et al.'s model was calculated by adding an *I* term to Fitts' law. The effect of *I* for the movement time was indicated only by the *I* term. We investigated fitting Zhang et al.'s model with the *I* term from Usuba et al.'s model (Eq. 8), and we found the highest adj. $R^2$ and lowest *AIC*.

$$MT = a + b_1 \log_2\left(\sqrt{c\left(\frac{A}{W}\right)^2 + (1-c)\left(\frac{A}{H}\right)^2} + 1\right) + b_2 \log_2\left(\frac{1}{I+0.0049} + 1\right) \quad (8)$$

Although Fitts' law can predict the movement time well, Eq. 8 can predict the target height *H* and interval *I* between the notch and target more accurately.

## 6 EXPERIMENT 2

In Experiment 1, we showed that the notch increases the pointing movement time under specific scenarios. Further, we found that participants moved the cursor based on two main strategies when the notch is placed between the start area and the target: (i) to move the cursor along the edge (along-strategy) and (ii) to avoid the notch (avoid-strategy). In Experiment 2, we investigated which of the above strategies is preferable in the current specification that allows the cursor to enter the notch.

### 6.1 Apparatus

We used a different apparatus for both experiments; however this did not have a significant effect on the conclusions of this study. We used a desktop PC (Intel Core i9-12900KF, GeForce RTX 3070 Ti, 32 GB RAM, Windows 10 Home). The display was manufactured by AOPEN (model 25XV2QFbmiiprx; 24.5" diagonal, 1920×1080 pixels) and its refresh rate was set at 360 Hz. We used an optical mouse (Logitech gaming mouse, G300s; 1600 DPI, and the mouse-cursor speed based on the OS setting was set to the middle of the slider in the control display and the "*Enhance pointer precision*" setting was turned on to match the usual settings of the participant.). The experimental system was implemented with Hot Soup Processor 3.6 and used in the full-screen mode.

### 6.2 Participants

A total of 12 local university students from a different participants group from that in Experiment 1 participated in this experiment. The average age was 22.3 years ($SD = 1.67$). All participants were skilled in mouse operation and used their dominant hand (right hand).

### 6.3 Task

Figure 6 shows a schematic of the task. We did not set the starting position of the trial as a condition, unlike that in Experiment 1. The starting area was a rectangle, and the trials started by simply clicking an area once. Except for this change, the task was the same as that in Experiment 1.

### 6.4 Design

We did not use *Position* and *Start* from Experiment 1 in Experiment 2, and we always placed the notch between the start area and the target. Instead, we added the condition of cursor movement strategy (*Strategy*). *Strategy = along* means moving the cursor along the top edge of the screen (Figure 7 (i)). *Strategy = avoid* means moving the cursor by avoiding the notch (Figure 7 (ii)). The target was a rectangle with a height of 6 mm (22 pixels) and a width (*W*) of 6 and 23 mm (22 and 84 pixels). The interval (*I*) between the notch and target was 0, 12, ∞ mm (0, 44, ∞ pixels). We chose the characteristic conditions from Experiment 1 for *W* and *I* to avoid increasing the workload on the participants, and we used almost equivalent values. The notch size and *A* were the same as those in Experiment 1.

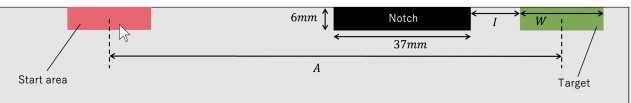

Figure 6: Schematic of Experiment 2.

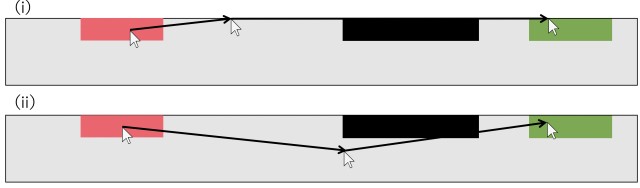

Figure 7: Comparison of *Strategy*. (i) Strategy that moves the cursor along the edge *along*. (ii) Strategy that avoids the notch *avoid*.

### 6.5 Procedure

One *set* comprised a random ordering of $2(A) \times 2(W) \times 3(I) = 12$ conditions. The participants performed the experiment for each *Strategy*. The participants first received an explanation about one *Strategy*, and they practiced employing this strategy in 24 trials (2 sets). Then, the participants performed 20 sets (240 trials) for data collection. After completing 20 sets, the participants took a 2 min break. After the break, the participants received an explanation for the other *Strategy*, which was practiced and performed similarly. Six of the participants started the experiment from *Strategy = along*, and the remaining six started from *Strategy = avoid*. Each participant took approximately 20 min to complete this experiment. We instructed participants using the same instructions as in Experiment 1. No participant performed a clutching action in this experiment as well.

### 6.6 Measurements

*MT* represents the time from clicking the start area to clicking the target, excluding the trials with errors. *ER* represents the percentage of clicks made outside the targets.

## 7 RESULT OF EXPERIMENT 2

We recorded 5,760 ($2(A) \times 2(W) \times 3(I) \times 2(Strategy) \times 20(set) \times 12(participants)$) trials including 241 error trials ($ER = 4.18\%$). This error rate was close to that in the previous study [28] (3.54%) and Experiment 1 (3.91%). We analyzed only error-free trials by *mean-of-means* calculation via repeated-measures ANOVA with a Bonferroni post-hoc test.

### 7.1 Error Rate (*ER*)

We observed the main effects of *W* ($F_{1,11} = 61.4$, $p < 0.001$, $\eta_p^2 = 0.848$) (Figure 8 (i)). The pair-wise comparisons showed that the error rates increased with a decrease in *W*. The other parameters did not show the main effects. We observed a significant interaction of $A \times Strategy$ ($F_{1,11} = 13.7$, $p = 0.00352$, $\eta_p^2 = 0.554$) (Figure 8 (i)). However, both error rates were close to those reported in the previous study [28].

### 7.2 Movement Time (*MT*)

Figure 8 (ii) shows the results of *MT*. We observed the main effect of *A* ($F_{1,11} = 330$, $p < 0.001$, $\eta_p^2 = 0.968$), *W* ($F_{1,11} = 404$, $p < 0.001$, $\eta_p^2 = 0.973$), *I* ($F_{2,22} = 87.2$, $p < 0.001$, $\eta_p^2 = 0.888$), and *Strategy* ($F_{1,11} = 24.1$, $p < 0.001$, $\eta_p^2 = 0.687$). The pair-wise comparisons showed that the *MT* for *Strategy = avoid* was shorter than that for *Strategy = along* ($p < 0.001$). Furthermore, the *MT* increased with

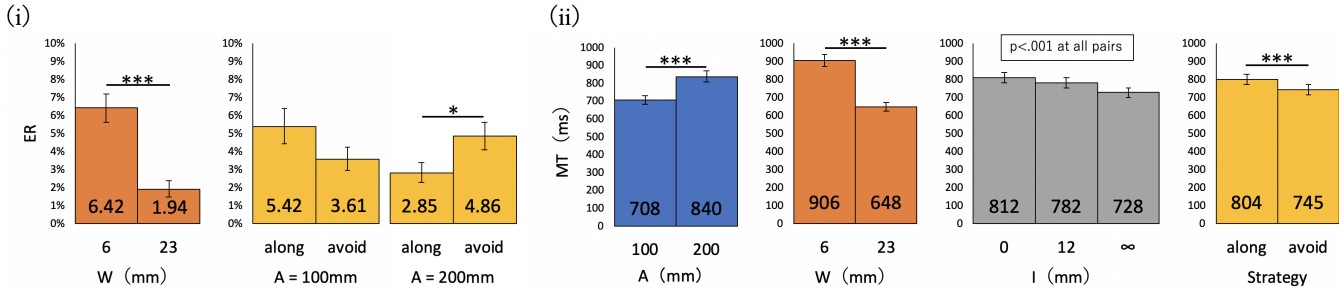

Figure 8: (i) Results of *ER* for *W* and *A* × *Strategy*. (ii) Results of *MT* for each parameter.

an increase in *A* ($p < 0.001$), decrease in *W* ($p < 0.001$), and decrease in *I* ($p < 0.001$ for all pairs). The effects of *A*, *W*, and *I* for *MT* were similar to those in Experiment 1. Although Experiment 2 used a different apparatuses and participants from those in Experiment 1, the results strongly corroborated with those of Experiment 1. We observed significant interactions of $W \times Strategy$ ($F_{1,11} = 69.1$, $p < 0.001$, $\eta_p^2 = 0.863$), $I \times Strategy$ ($F_{2,22} = 15.5$, $p < 0.001$, $\eta_p^2 = 0.584$), $A \times W \times Strategy$ ($F_{1,11} = 8.29$, $p = 0.0150$, $\eta_p^2 = 0.430$), $A \times I \times Strategy$ ($F_{2,22} = 5.28$, $p = 0.0134$, $\eta_p^2 = 0.324$), and $W \times I \times Strategy$ ($F_{2,22} = 9.67$, $p < 0.001$, $\eta_p^2 = 0.468$).

### 7.3 Virtual Travel Distance

We analyzed the virtual travel distance in *PE* to discuss the distance the mouse moved. We observed the main effects of *A* ($F_{1,11} = 177 \times 10$, $p < 0.001$, $\eta_p^2 = 0.994$), *W* ($F_{1,11} = 43.4$, $p < 0.001$, $\eta_p^2 = 0.798$), *I* ($F_{2,22} = 24.6$, $p < 0.001$, $\eta_p^2 = 0.691$). *Strategy* did not show the main effect ($F_{1,11} = 2.49$, $p = 0.143$, $\eta_p^2 = 0.185$). Although there was no main effect, *Strategy* = *avoid* was always longer than *Strategy* = *along*. We observed significant interactions of $A \times Strategy$ ($F_{1,11} = 4.91$, $p = 0.0487$, $\eta_p^2 = 0.309$), $I \times Strategy$ ($F_{2,22} = 4.08$, $p = 0.0312$, $\eta_p^2 = 0.270$), $A \times I \times Strategy$ ($F_{2,22} = 3.78$, $p = 0.0387$, $\eta_p^2 = 0.256$).

## 8 DISCUSSION OF EXPERIMENT 2

### 8.1 Movement Time

*Strategy* = *avoid* tends to have a longer virtual travel distance than *Strategy* = *along* and a shorter movement time. This result indicates that it is desirable to employ *Strategy* = *avoid*, although it increases the distance required to move the mouse. For *A* = 100, *Strategy* = *avoid* has a shorter *MT* than that of *Strategy* = *along* at *W* = 23; however, the difference is not significant. This suggests that the effect of *Strategy* is smaller when *ID* in Eq. 1 is small.

From the significant interaction of $A \times I \times Strategy$ and $W \times I \times Strategy$, we observed that the effect of *I* for the *MT* was small for *Strategy* = *avoid*. In *Strategy* = *avoid*, the participants aimed at the target from the lower side after avoiding the notch. Therefore, we believe the effect of *I* to be smaller. In Experiment 1, we showed that placing the target near the notch increased *MT*. However, if the user moved the cursor using the avoid-strategy, the effect of the interval between the notch and the target for the movement time would probably decrease.

### 8.2 User Strategy

Ten participants answered that they preferred *Strategy* = *avoid*. They reported some stress when using *Strategy* = *along*, which caused them to lose sight of the cursor in the notch or to pass far past the notch. They prefer *Strategy* = *avoid* because it did not cause such stress. One participant had a shorter average movement time

Table 2: Model fitting for candidate models.

| Strategy | Model | a | $b_1$ | c | $b_2$ | adj. $R^2$ | AIC |
|----------|-------|-----|-------|-------|------|-----------|------|
| both | | 239 | 46.3 | | | 0.866 | 149 |
| along | Eq. 1 | 199 | 162 | | | 0.877 | 71.2 |
| avoid | | 279 | 126 | | | 0.966 | 61.5 |
| both | | 193 | 144 | 22.7 | | 0.910 | 131 |
| along | Eq. 5 | 133 | 162 | 33.2 | | 0.964 | 66.1 |
| avoid | | 277 | 126 | 0.678 | | 0.986 | 58.1 |
| both | | 229 | 145 | | | 0.878 | 133 |
| along | Eq. 6 | 186 | 164 | | | 0.894 | 70.4 |
| avoid | | 271 | 126 | | | 0.973 | 60.3 |
| both | | 219 | 144 | 1.00 | 7.47 | 0.890 | 135 |
| along | Eq. 8 | 170 | 162 | 1.00 | 11.2 | 0.929 | 71.0 |
| avoid | | 229 | 134 | 0.982 | 3.72 | 0.975 | 62.6 |

for *Strategy* = *along* than that for *Strategy* = *avoid*; however, the difference was small (7 ms) and they preferred *Strategy* = *avoid*. The other two participants reported that they preferred *Strategy* = *along* because they only needed to move the cursor in the left and right directions. The average movement time was shorter for *Strategy* = *avoid* than for *Strategy* = *along* in both participants.

### 8.3 Model Fitting

Table 2 lists the results of model fitting. In column *Strategy*, *both* shows the results of model fits using all 24 data points; *along* and *avoid* show the results of model fitting using 12 data points in each *Strategy*.

Fitts' law (Eq. 1) showed good fits above the typical threshold ($R^2 > 0.9$) only for *avoid* and low fits for *Strategy* = *both* that differed from those of Experiment 1 and the previous study [28]. This can be attributed to the fact that the strategy of movement was clearly specified in Experiment 2.

We investigated the fits to the model for the case when there is a distractor on the path to the target and when the user avoids the distractor (Eq. 5 and Eq. 6). We set *B* in Eq. 5 and Eq.6 to 3 mm, which is half the height of the notch placed in Experiment 2. Furthermore, we investigated the fits to the proposed model (Eq. 8). Eq. 5 showed the highest adj. $R^2$ and lowest *AIC* in *Strategy* = *avoid* because Eq. 5 assumes obstacle avoidance. Eq. 5 showed the highest adj. $R^2$ and lowest *AIC* for all *Strategy*s. Vaughan et al. pointed out that Eq. 5 simply adds a *B* term to Fitts' law and lacks a mechanistic account. In addition, Eqs. 5 and 6 do not consider the effect of *I*. Therefore, we believe that Eq. 8 deserves consideration as a model in Experiment 2 because it shows no significant difference in *AIC* relative to the other models and shows good fits above the typical threshold ($R^2 > 0.9$) for *Strategy* = *along* and *Strategy* = *avoid*.

## 9 EXPERIMENT 3

Changing the notch to an area where the cursor cannot enter can be considered as an effective approach as the avoid-strategy was found to be desirable in Experiment 2. Experiment 3 was almost the same experiment as Experiment 2; however, the notch was changed to an

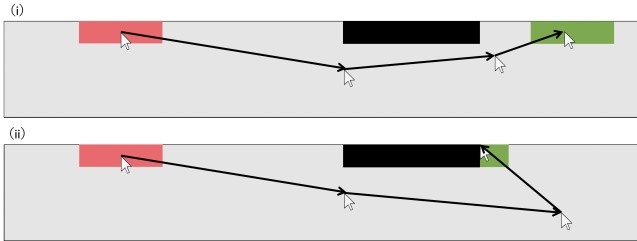

Figure 9: Overview of the strategy dictated to the participants based on the interval between the notch and target in Experiment 3. (i) A condition where the notch and target are not adjacent. We instructed participants to avoid the notch. (ii) A condition where the notch and target are adjacent to each other. We instructed participants to move the cursor by stopping on the outer edge of the notch.

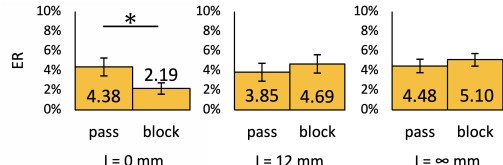

Figure 10: Results of *ER* for $I \times Notch$.

area where the cursor cannot enter. The apparatuses, participants, task, and measurements were the same as in Experiment 2.

### 9.1 Design

The notch was in an area that the cursor cannot enter. We did not use *Strategy* from Experiment 2 in Experiment 3. We set the notch condition (*Notch*) as an experimental condition. *Notch = block* indicates the cursor cannot enter the notch. *Notch = pass* implies the cursor can enter the notch (using the data of *Strategy = avoid* in Experiment 2). We instructed participants to (1) avoid the notch if the condition where the notch and the target are not adjacent (Figure 9 (i)) and (2) move the cursor by stopping on the outer edge of the notch if the condition where the notch and target are adjacent (Figure 9 (ii)). Other conditions were the same as those in Experiment 2.

### 9.2 Procedure

One *set* comprised a random ordering of $2(A) \times 2(W) \times 3(I) = 12$ conditions. Participants first practiced 48 trials (4 sets) to become familiar with the movement shown in Figure 9. Then, the participants performed 20 sets (240 trials) for data collection. Each participant took approximately 10 min to complete this experiment, and the same instruction as in Experiment 2 were provided. No participant performed a clutching action in this experiment as well.

## 10 RESULTS OF EXPERIMENT 3

We recorded 5,760 ($2(A) \times 2(W) \times 3(I) \times 2(Notch) \times 20(set) \times 12(participants)$) trials including 237 error trials ($ER = 4.11\%$). This error rate was close to that in previous study [28] (3.54%), Experiment 1 (3.91%), and Experiment 2 (4.18%). We analyzed only error-free trials by *mean-of-means* calculation via repeated-measures ANOVA with a Bonferroni post-hoc test.

### 10.1 Error Rate (*ER*)

We observed the main effects of $W$ ($F_{1,11} = 72.1$, $p < 0.001$, $\eta_p^2 = 0.868$). The pair-wise comparisons showed that error rates increased with a decrease in $W$. The other parameters did not show the main effects. We observed a significant interaction of $I \times Notch$ ($F_{2,22} =$

3.91, $p = 0.0352$, $\eta_p^2 = 0.262$) (Figure 10). For $I = 0$, $Notch = block$ was significantly lower than $Notch = pass$.

### 10.2 Movement Time (*MT*)

We observed the main effects of $A$ ($F_{1,11} = 443$, $p < 0.001$, $\eta_p^2 = 0.976$), $W$ ($F_{1,11} = 417$, $p < 0.001$, $\eta_p^2 = 0.974$), $I$ ($F_{2,22} = 21.8$, $p < 0.001$, $\eta_p^2 = 0.664$), and *Notch* ($F_{1,11} = 10.8$, $p = 0.00718$, $\eta_p^2 = 0.496$). These results showed that *MT* increased with an increase in $A$ ($p < 0.001$), decrease in $W$ ($p < 0.001$), and decrease in $I$ ($I = 0 \times I = 12$ ($p < 0.001$), $I = 0 \times I = \infty$ ($p = 0.0169$), and $I = 12 \times I = \infty$ ($p = 0.0276$)). We observed significant interactions of $A \times W$ ($F_{1,11} = 8.36$, $p = 0.0147$, $\eta_p^2 = 0.432$), $A \times I$ ($F_{2,22} = 9.02$, $p = 0.00138$, $\eta_p^2 = 0.450$), $W \times I$ ($F_{2,22} = 41.9$, $p < 0.001$, $\eta_p^2 = 0.792$), $W \times Notch$ ($F_{1,11} = 27.9$, $p < 0.001$, $\eta_p^2 = 0.717$), $I \times Notch$ ($F_{2,22} = 57.0$, $p < 0.001$, $\eta_p^2 = 0.838$), $A \times I \times Notch$ ($F_{2,22} = 3.88$, $p = 0.0359$, $\eta_p^2 = 0.261$), and $W \times I \times Notch$ ($F_{2,22} = 37.1$, $p < 0.001$, $\eta_p^2 = 0.771$). The significant interaction of $A \times I \times Strategy$ and $W \times I \times Strategy$ indicates that $Notch = block$ is significantly shorter than $Notch = pass$ at $I = 0$ (Figure 11).

### 10.3 Virtual Travel Distance

We observed the main effects of $A$ ($F_{1,11} = 578$, $p < 0.001$, $\eta_p^2 = 0.981$), $W$ ($F_{1,11} = 14.7$, $p = 0.00274$, $\eta_p^2 = 0.573$), $I$ ($F_{2,22} = 23.4$, $p < 0.001$, $\eta_p^2 = 0.681$), and *Notch* ($F_{1,11} = 22.0$, $p < 0.001$, $\eta_p^2 = 0.667$). We observed significant interactions of $A \times I$ ($F_{2,22} = 6.77$, $p = 0.00512$, $\eta_p^2 = 0.381$), $A \times Notch$ ($F_{1,11} = 8.59$, $p = 0.0137$, $\eta_p^2 = 0.438$), $W \times I$ ($F_{2,22} = 4.45$, $p = 0.0239$, $\eta_p^2 = 0.288$), $I \times Notch$ ($F_{2,22} = 19.0$, $p < 0.001$, $\eta_p^2 = 0.633$), $A \times I \times Notch$ ($F_{2,22} = 9.43$, $p = 0.00110$, $\eta_p^2 = 0.462$ (Figure 12 (top))), and $W \times I \times Notch$ ($F_{2,22} = 4.53$, $p = 0.0226$, $\eta_p^2 = 0.292$ (Figure 12 (bottom))). The significant interaction of $A \times I \times Strategy$ and $W \times I \times Strategy$ indicates that $Notch = block$ is significantly longer than $Notch = pass$ at $I = 0$. (Figure 12).

### 10.4 Path Efficiency *PE*

We observed the main effects of $A$ ($F_{1,11} = 21.3$, $p < 0.001$, $\eta_p^2 = 0.659$), $W$ ($F_{1,11} = 5.36$, $p = 0.0410$, $\eta_p^2 = 0.328$), $I$ ($F_{2,22} = 23.0$, $p < 0.001$, $\eta_p^2 = 0.676$), and *Notch* ($F_{1,11} = 20.5$, $p < 0.001$, $\eta_p^2 = 0.650$). We observed significant interactions of $A \times I$ ($F_{2,22} = 14.1$, $p < 0.001$, $\eta_p^2 = 0.562$), $A \times Notch$ ($F_{1,11} = 13.1$, $p = 0.00406$, $\eta_p^2 = 0.543$), $W \times I$ ($F_{2,22} = 3.50$, $p = 0.0481$, $\eta_p^2 = 0.241$), $I \times Notch$ ($F_{2,22} = 32.7$, $p < 0.001$, $\eta_p^2 = 0.748$), $A \times I \times Notch$ ($F_{2,22} = 10.8$, $p < 0.001$, $\eta_p^2 = 0.496$ (Figure 13 (top))), and $W \times I \times Notch$ ($F_{2,22} = 5.54$, $p = 0.0113$, $\eta_p^2 = 0.335$ (Figure 13 (bottom))). The significant interaction of $A \times I \times Strategy$ and $W \times I \times Strategy$ indicates that $Notch = block$ is significantly lower than $Notch = pass$ at $I = 0$ (Figure 13).

## 11 DISCUSSION OF EXPERIMENT 3

### 11.1 Movement Time (*MT*)

*Notch = block* had a significantly shorter *MT* than *Notch = pass* at $I = 0$ (Figure 11). For $I = 0$, *Notch = block*'s long virtual travel distance and low *PE* showed that participants moved the cursor by stopping at the outer edge of the notch (Figure 14). Therefore, we believe that the participants pointed the target in a short movement time without precise control. In addition, there was a significant difference in movement time only at $I = 0$. Therefore, we concluded that changing the notch to an area where the cursor cannot enter does not change the movement time of the target that is not adjacent

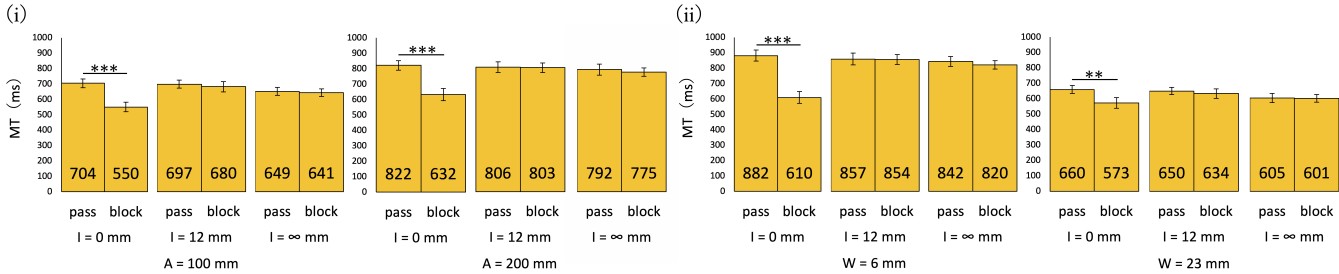

Figure 11: Results of *MT* for (i) $A \times I \times Notch$ (ii) $W \times I \times Notch$.

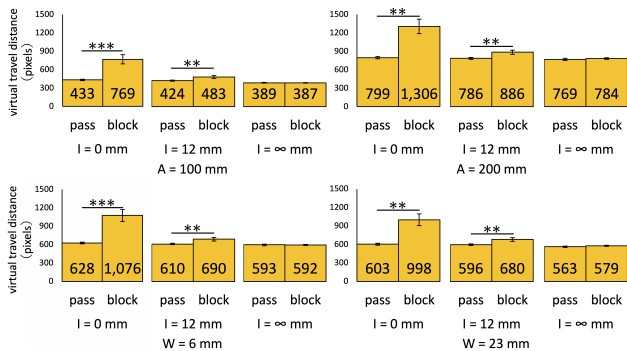

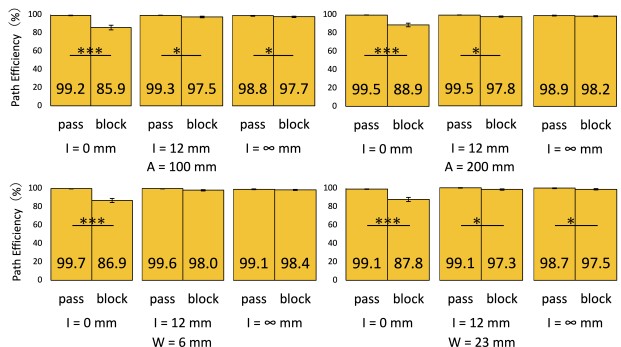

Figure 12: Results of virtual travel distance for (top) $A \times I \times Notch$ (bottom) $W \times I \times Notch$.

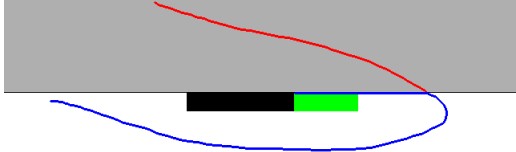

Figure 13: Results of *PE* for (top) $A \times I \times Notch$ (bottom) $W \times I \times Notch$.

to the notch. It only reduces the movement time for a target adjacent to the notch.

## 11.2 Error Rate (*ER*)

For $I = 0$, $Notch = block$ was significantly lower than that for $Notch = pass$. We believe that this result is attributed to cursor stops on the outer edge of the notch and can be pointed without precise control, as is the case with the target adjacent to the edge of the screen (Figure 14).

## 11.3 Participants Questionnaire

All participants answered that they preferred the notch through which the cursor cannot enter. Furthermore, they preferred conditions where the notch and target were adjacent and easier to move because the cursor stopped on the notch. No participant answered that the physical workload of the movement increased, although the virtual travel distance did increase. Three participants reported that the cursor was sometimes unnecessarily caught in the notch.

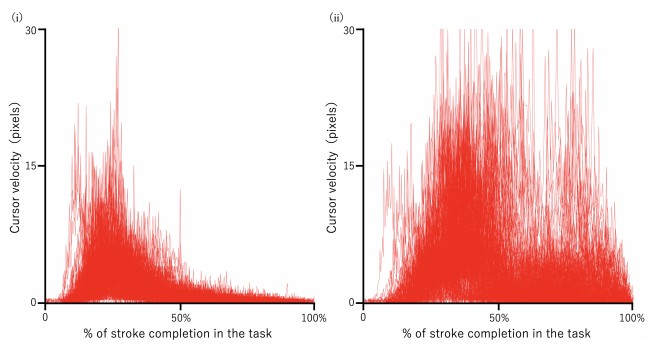

Figure 14: Example cursor trajectory. The red lines indicate the virtual cursor trajectories assuming there is no screen edge; the blue lines indicate the actual on-screen trajectories.

Figure 15: Cursor velocity against the percentage of elapsed time in (i) $Notch = pass$ (ii) $Notch = block$.

## 11.4 Cursor Velocity

Figure 15 shows the cursor velocity against the percentage of elapsed time in the task. Figure 15 shows the condition $A = 200$, $W = 6$, and $I = 0$. In pointing tasks, there is a peak in velocity at the beginning of the movement, and it slows down when the cursor moves inside the target [4]. Figure 15 (i), which shows the cursor velocity for $Notch = pass$, shows the same trend as that in the previous study [4]. In contrast, Figure 15 (ii), which shows the cursor velocity for $Notch = block$, shows that it does not slow down considerably in the second half. This trend was similar to that of Yamanaka et al. [27] in their study of the cursor velocity of targets adjacent to the edge of the screen considering that the cursor stops on the outer edge of the screen. This allows the user to point at the target while moving the cursor at high speed.

## 12  GENERAL DISCUSSION

We showed that the notch increased movement time in specific situations by hiding the cursor in Experiment 1. The fact that participants avoided the notch or moved the mouse cursor along the top edge of the screen may have influenced the increased movement time. The notch caused the participant to miss the cursor position inside the

notch or lose sight of the cursor. We also found that it is better to avoid the notch instead of moving the cursor along the top edge in the current specification of the notch in Experiment 2.

In addition, we showed that the effect of the interval between the notch and the target for movement time reduced if the user moved the cursor by avoiding the notch. Most participants preferred the strategy of avoiding the notch, and although the distance to move the mouse was longer, the movement time was shorter.

In Experiment 3, we found that changing the notch to an area where the cursor cannot enter was found to be effective for pointing at the target faster and more accurately when the target was adjacent to the notch. We believe that the participants pointed the target in a short movement time without precise control and considering that the target can be pointed simply by hitting the outer edge of the notch. The results of the cursor velocity and *PE* supported these arguments. In addition, there was no significant difference in movement time when the target was not adjacent because of the change in the notch specification.

Therefore, changing the notch to an area where the cursor cannot enter eliminates the situation that the cursor is hidden by the notch, which allows a faster and more accurate pointing the target which adjacent to the notch.

## 13 LIMITATION AND FUTURE WORK

We set the size of the notch in this study based on the size of the notch on the MacBook Pro (2021). However, the notch size may change in future products. In addition, a larger cursor size would eliminate the situation where the cursor is completely hidden by the notch. In such cases, a strategy of moving the cursor along the top edge of the screen may be preferred over a strategy of avoiding the notch. In the future, we want to study the notch size and cursor size along the experimental conditions.

In the experiments of this study, we placed all targets adjacent to only one edge of the screen. We did not investigate the target on the corner of the screen (i.e., the target adjacent to the two screen edges). *MT* depends only on *A* if *ID* in Eq. 1 is small [12] (Eq. 9).

$$ID = \sqrt{A} \qquad (9)$$

The movement time may be predicted by Eq. 9 because the corner target can be pointed to without precise control. Therefore, a limitation of this study is that the effect of the notch for the corner target is not investigated.

The model proposed in this study (Eq. 8) showed a good fit to the present experimental conditions. We believe that our model (Eq. 8) can be applied to predict the movement time of methods that require a continuous cursor movement while avoiding distractors [29] (e.g., Bubble Clusters [26] and Attribute Gates [24]). We are interested in whether Eq. 8 can be applied to more than just predicting movement time in the scenario where the notch is placed.

We used a general cursor in all our experiments. However, pointing-facilitation technique have been proposed to improve the pointing performance. For example, the notch effect for the movement time may be reduced when using Bubble Cursor [13] or Ninja Cursors [18]. However, we did not consider them in this study because of the somewhat tricky behavior of these techniques.

We instructed the participants to avoid any clutching action during the trial; however, in actual use of the PC, the user may perform clutching actions. For example, if the cursor is hidden by a notch during a clutching action, the effect of the notch may be increased. Restricting the clutch action may have restricted the user's operation strategy, which is a limitation of this study.

In this study, we only considered the movement time and error rate for evaluation. However, placing the notch can increase the psychological stress experienced by users. For example, the user may feel uncomfortable about the cursor unnecessarily getting caught in the notch if the notch is changed to an area where the cursor cannot enter. Therefore, we only investigated the effect of the notch for movement time and error rate, which is another limitation.

## 14 CONCLUSION

Based on a series of experiments, this study suggested that the notch should be an area where the cursor cannot enter considering that the cursor stops at the outer edge of the notch, which allows the fast pointing of a target adjacent to the notch. Experiment 1 showed that the notch increased the pointing movement time in specific scenarios. Experiment 2 indicated that it is better to avoid the notch than enter the notch given the current specification of the notch. Experiment 3 showed that changing the notch to an area where the cursor cannot enter is effective for pointing the target faster and more accurately if the target is adjacent to the notch. The cursor velocities indicated participants pointed at a target adjacent to the notch with high speed. Therefore, **the notch should be an area where the cursor cannot enter**.

We offer the following contributions:

- We showed that the notch that hides the cursor increases movement time. Further, we presented a preferred operation strategy when the notch is placed on the path to the target. We believe that MacBook Pro (2021) users should move the mouse cursor according to this strategy.

- Changing the notch to an area where the cursor cannot enter allows for faster and more accurate pointing of the target adjacent to the notch. We presented one indicator of a suitable specification for placing a notch on a display.

### ACKNOWLEDGMENTS

We would like to thank Editage (www.editage.com) for English language editing.

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
