# OpenReview forum: "Effect of the Notch a Cursor Cannot Enter for Pointing Movement Time"
_graphicsinterface.org/Graphics_Interface/2023/Conference — GI 2023_

### Official Review · Reviewer_WQr7 · 2023-01-13
**Slightly positive**

**Rating:** 7
**Confidence:** 4

**Review:**

In this submission, the authors set out to assess the impact of the MacBook's Notch on user pointing tasks. The paper contains three different user studies that aim at assessing different strategies or the impact of different parameters that the notch can have on pointing tasks.

The paper is very relevant to the conference and its attendees and is overall quite easy to read. That being said, while the authors thank a company for english editing, I still found some sentences that I struggled to grasp or understand and I wonder if some more efforts could not be spent on the writing of the manuscript to get the message across in an easier fashion. I am not a native speaker myself nor am I very good at exactly finding or pointing out to errors, but some sentences in the manuscript definitely read as quite bizarre to me.

The paper makes a lot of contributions and its length is appropriate considering those. Although the authors seem to tackle a problem that is, one might argue, from a very isolated case (being based on the design of a single laptop), the findings can be useful to a large branch of the community and industrials as well so I would not argue against publication based solely on this fact.

The authors seem to have used statistical tests without verifying the assumptions that they have. Could the authors justify that the assumptions of these tests are not violated?
I wonder if the authors could provide exact p-values as well, at least for p-values that are greater than 0.001. Indeed, I see for instance a p < 0.05 on page 3 for which readers could benefit from an exact value (see discussions by Cockburn et al. ). This notation "(p < 0.001 to 0.05)" in 5.1 is also very difficult to understand for me.

Finally, I wonder if the authors would be willing to sharing their code for these experiments such that they could be replicated or transferred into other applications and future research.

Overall, I am quite positive about the submission. I would argue that it's a good fit for the conference and that the paper is probably robust enough should my concerns about the statistical analysis and the tests used.


Reference:
Cockburn et al. https://doi.org/10.1145/3360311

---

### Official Review · Reviewer_8kQ3 · 2023-01-13
**Experiements were overall well-done, I only have some minor suggestions for presentation clarity.**

**Rating:** 7
**Confidence:** 4

**Review:**

The authors presented in this manuscript three studies to investigate human performance of moving the mouse cursor with a notch area presented on the screen,  Overall, I think this manuscript is clearly written and the work is well done. It is a pleasure for me to read. I do not have grave concerns but a few points that I would expect more clarification/improvements.

The authors clearly presented the motivation in the introduction, I only have a minor comment: it might be clearer if paragraph 3 was put after paragraphs 4/5 as it is kind of the results of the studies. I think it might be natural to first describe what has been done and then list the results.

The related work is well structured, but I think it lacks some discussions, like what are the limitations of the previous work and how previous work inspires or is different from this one.

In study 1,  most parameters were described in mm, but the diameter of the start area was described in pixels. Although we could manually convert them with the parameters of the experimental apparatus, I still think it might be better to unify them (or present in both pixel and mm).

It would be nice to add some more details for the design of the study because we expect one paper should be understandable without the need to read too many previous works. For example, in addition to "We set the I values with reference to the previous study [23]", it would be nice to briefly summarize the reasons,

I understand the reason for asking the participants to avoid clutching actions, I wonder since there are no hard rules to disable them from doing it in Study 1, how these rules were followed exactly during the studies. Also, I think it would be nice to add a discussion of this rule to the realism of this study because in real-world scenarios, users might just clutch it for speed.

For all studies, I understand the authors' intention to fit theoretical models with their measured data, but it seems that the usefulness of the models was not sufficiently discussed, thus it is not trivial to know what are the add-values of fitting the models. The authors could for example extend a little bit to give some real-world examples of how the models could be used in the future world.

For Study 2, I did not get if the order of Strategy was fixed, counter-balanced, or randomized for each participant.

The plot of results could have some space to improve, such as adjusting the width of each bar and the distance between them. Also, the authors might consider stating what the error bars represent in their figure descriptions.

---

### Official Review · Reviewer_qNPz · 2023-01-20
**Focused but in-depth study that probably should be published**

**Rating:** 7
**Confidence:** 3

**Review:**

This paper presents three studies looking at Fitts's Law style studies analyzing the effect of a notch, like that of the Macbook Pro (2021), has on certain nearby targets and certain trajectories. The high-level conclusion is that the notch should not be a passable region in this case.

Review: I think this paper should probably be accepted. While it is a focused contribution, it is an interesting finding and a substantial, in-depth analysis. For the most part, the methods are sound, although I have some comments about the experiments (see below). The final conclusions seem reasonable to me given the experiments, and there is a wealth of results for readers to draw from.

Comments:
 - I wonder if just doing experiment three would be a more elegant, easy-to-read contribution. Experiment 1 suggests that the notch makes a difference, but it's a little indirect (as it is inferred from the distance of the notch being a significant factor). Experiment 2, which relies on participants following a prescribed strategy, is interesting but means that the strategy isn't as controlled. However, there is a wealth of information here, so I think including all studies is good.
- The statistics and graphs can be overwhelming at times. In particular, the discussion of relevant equations is a deep dive. However, the reasoning and statistics seem sound to me at all levels. (I am less expert on the typical methods of Fitts's Law studies, so I defer to other reviewers there.)
- Related work seems a little light, but it does not seem to be a problem for me
- A careful read-through should be conducted (see below)


Minor writing points:
- The focus on the Macbook Pro (2021) is a little strange as overall motivation, I wonder if notches should be described generally and the Macbook Pro (2021) cited as an example that this paper explores. That said, I think it's fine, it's just a stylistic quirk.
 - The description of experiment 1 results in the second paragraph of the intro is strange to me.
 - As stated, experiment one doesn't directly look at the effect of the presence of the notch, but rather the distance of the target from the notch.
 - typo "minfor" page 3
 - should it be a threshold of r^2 < 0.9 at the end of section 8.3, or > 0.9?

---

### Meta-Review · Area_Chair_eJdR · 2023-01-22

**Recommendation:** 9
**Confidence:** 5

**Metareview:**

In this manuscript the authors present three studies looking at Fitts's Law style studies analyzing the effect of a notch with the example of the notch in the MacBook Pro.

All reviewers agree that the manuscript is very well written (although they all recommend minor improvements or suggestions here and there that I want to encourage the authors to look at) and that the contribution although very much in-depth, warrants a publication.
For this reason, I am happy to recommend acceptance of the manuscript.

I would like to encourage the authors to revise their work to include some of the suggestions from the reviewers, including mostly, the presentation of results, either graphical or textual. That being said, the authors should probably look at all suggestions to revise their work for a very minor round of revisions before the camera-ready version.